# The Contribution of Digital Sequence Information to Conservation Biology: A Southern African Perspective

Isa-Rita M. Russo,* Deon de Jager, Anna M. van Wyk, Arrie W. Klopper, Kenneth Uiseb, Coral Birss, Ian Rushworth, and Paulette Bloomer*

Many recent contributions have made a compelling case that genetic diversity is not adequately reflected in international frameworks and policies, as well as in local governmental processes implementing such frameworks. Using digital sequence information (DSI) and other publicly available data is supported to assess genetic diversity, toward formulation of practical actions for long-term conservation of biodiversity, with the particular goal of maintaining ecological and evolutionary processes. Given the inclusion of specific goals and targets regarding DSI in the latest draft of the Global Biodiversity Framework negotiated at the 15th Conference of the Parties (COP15) in Montreal in December 2022 and the crucial decisions on access and benefit sharing to DSI that will be taken in the coming months and future COP meetings, a southern African perspective on how and why open access to DSI is essential for the conservation of intraspecific biodiversity (genetic diversity and structure) across country borders is provided.

I.-R. M. Russo
School of Biosciences
Cardiff University
Cardiff CF10 3AX, UK
E-mail: russoim@cardiff.ac.uk

D. de Jager
Section of Molecular Ecology and Evolution
Globe Institute
University of Copenhagen
Copenhagen 1353, Denmark

A. M. van Wyk, A. W. Klopper, P. Bloomer
Department of Biochemistry
Genetics and Microbiology
University of Pretoria
Pretoria 0002, South Africa
E-mail: paulette.bloomer@up.ac.za

K. Uiseb
Ministry of Environment, Forestry and Tourism
Windhoek 13306, Namibia

C. Birss
CapeNature
Cape Town 7766, South Africa

I. Rushworth
Ezemvelo KZN Wildlife
Pietermaritzburg 3201, South Africa

The fundamental components of biodiversity are ecosystem, species, and genetic diversity.[1] Genetic diversity is essential to evolutionary processes and is important in ecosystem stability, while low genetic diversity increases the extinction risk of populations.[2,3] This aspect of biodiversity, however, is often underrepresented in national and international policy dealing with conservation and management of ecosystems and species.[4] This oversight may lead to management tools being applied inappropriately from an evolutionary or genetic perspective, with potential negative consequences for species and ecosystems.[5]

One such key management and conservation tool, globally, is translocation. Between 130 000 and 170 000 animals are estimated to be translocated annually in South Africa alone, predominantly within the billion-dollar wildlife ranching industry, but also within the national and provincial protected area network.[6–8] The wildlife ranching industry in Namibia is growing rapidly, and many animals are translocated between southern African countries, and further afield, for conservation purposes.[9,10] Between 1973 and 1989, more than 700 translocation events took place per year in the USA, Australia, and New Zealand.[11] The goal of translocations (moving individuals from one population to another) is often to maintain and/or increase genetic diversity in the receiving population. This allows connectivity (gene flow) between populations that have been isolated by human activities, such as land-use change, urbanization, and fences. Genetic diversity allows populations to adapt to changing conditions over time, including disease outbreaks and environmental changes. The implementation of translocation programs has been instrumental in some of conservation's most celebrated success stories such as the white rhinoceros (*Ceratotherium simum*), Arabian oryx (*Oryx leucoryx*), the California condor (*Gymnogyps californianus*), and genetic rescue of the Florida panther (*Puma concolor coryi*).[12–15]

Translocation, however, can also be a threat to wildlife, as it may promote unintended hybridization between closely related species, subspecies, or differentiated populations, potentially leading to a breakdown of evolutionary processes.[16] Consequently, extensive translocations in the absence of genetic information may threaten the genetic integrity of species and the long-term resilience of populations, species, and ecosystems. Furthermore, the implementation of translocations is often complex and

expensive, and the results are not always desirable.[17] Thus, all available information should be considered when translocations are implemented as a management tool, to maximize the chance of success and minimize the potential harm (including ecological and evolutionary harm).

In southern Africa, where translocation of wildlife is common, genetic information is seldom incorporated into the decision-making process around translocations. This is despite the availability of population- and conservation genetics studies and Digital Sequence Information (DSI), which is a policy term that refers to digitally stored information from DNA and RNA (i.e., genetic sequences). Some of the reasons for this include lack of access (due to a paywall) to the relevant literature for conservation managers and permit officials, lack of training in the interpretation of genetic findings, and the authors of such studies not explicitly and adequately stating the conservation or management implications of their findings. One aspect that is not prohibitive in this case, is access to DSI of the species under consideration via public databases (e.g., the databases of the International Nucleotide Sequence Database Collaboration, https://www.insdc.org/, and The Barcode of Life Data System, https://www.boldsystems.org).

Consequently, access to DSI is an essential resource for southern African researchers in promoting consideration of genetics in translocation policy and decisions. The main idea behind incorporating genetics into translocation decisions is to maintain patterns of genetic diversity and structure (genetic composition) within a species that were generated by evolutionary processes (as opposed to human-driven processes). These may include, but are not limited to, adaptation of a population to local environmental conditions, genetic differentiation due to the development of behavioral differences that prevent interbreeding between populations even though they might occur in sympatry or the genetic differentiation of populations due to a physical geographic barrier (e.g., a river, a mountain, uninhabitable environment). Many African wildlife species that are commonly translocated show such genetic differentiation patterns across the continent.[18] These patterns are often identified in phylogeographic studies based on genetic sequence data, such as mitochondrial DNA or nuclear genome sequences, of the species from various countries across its distribution range. In addition, publicly available microsatellite/single nucleotide polymorphisms (SNP) data (e.g., on DRYAD or Zenodo) can be used to investigate finer scale patterns of population differentiation. It is these patterns and DSI data (uploaded to open access databases) that can be leveraged to improve translocation decisions for the benefit of the species. For example, in lions (*Panthera leo*), researchers and practitioners have defined conservation units between which translocations should or should not be performed to conserve the evolutionary genetic units of the species.[19]

One way that we leverage DSI from published studies is to combine the genetic sequences available from GenBank with newly generated sequences (often from understudied regions) to obtain a more comprehensive view of the genetic composition of a species. A crucial point mentioned above is the fact that most translocated species occur in more than one African country. Therefore, existing DSI for a species from countries other than the one or two we might have generated new data from is essential to this endeavor. Without the context of DSI from other countries of the species' range, new genetic data generated will be almost meaningless. For example, for black (*Diceros bicornis*) and white (*C. simum*) rhinoceros, Moodley et al.[20] used two publicly available genome sequences (DSI) from one African and one Asian country, and three newly generated genomes from three African countries, to investigate intraspecific gene flow and the evolution of feeding specialization. Another example is for lion (*P. leo*), where Bertola et al.[21] used 114 publicly available sequences (DSI) from 22 different countries, and 91 newly generated sequences from 18 different countries to investigate the phylogeographic patterns and genetic clustering of African lions. This study, and the data used therein, was a necessary precursor to being able to define lion conservation units for conservation actions (translocation).[19] Thus, by using publicly available DSI, not only is our understanding of the evolution and genetic composition of a species improved, but new biodiversity knowledge is generated, both of which are crucial aspects for biodiversity conservation identified by the Convention on Biological Diversity (CBD). In turn, the new genetic data generated are openly accessible as DSI to other African researchers, to further their understanding of their local populations of a particular species.

Goal C in the latest Kunming-Montreal Global Biodiversity Framework (Draft decision submitted by the President, CBD/COP/15/L.25, published on 18 December 2022) states "The monetary and non-monetary benefits from the utilization of genetic resources and digital sequence information on genetic resources, … are shared fairly and equitably", while Target 13 actions "Take effective legal, policy, administrative and capacity-building measures…" to implement Goal C.[22] Furthermore, paragraph 16 of the document CBD/COP/15/L.30 (Digital sequence information on genetic resources) states that The Conference of the Parties (COP) "Decides to establish, as part of the post-2020 global biodiversity framework, a multilateral mechanism for benefit-sharing from the use of digital sequence information on genetic resources, including a global fund", and in paragraph 9 states that the mechanism should "Not hinder research and innovation" and "Be consistent with open access to data".[23] We support this multilateral, open access, and benefit-sharing option for DSI under the Kunming–Montreal Global Biodiversity Framework.[24] This solution will not only benefit researchers and conservation practitioners from southern Africa, but also those from neighboring countries that often share the inherently valuable wildlife species being translocated, as well as many other African countries where these species occur. A bilateral option, where access to DSI must be applied for and granted through a bureaucratic system, or where access has to be paid for, would have significantly delayed or completely ceased any genetic input to translocation decisions, or prevented any genetic data from being incorporated in the future, to the detriment of all countries involved and the very species we are trying to protect.

There have been few initiatives across the globe that have tried to standardize the incorporation of genetic data into translocation policy and decisions.[25] Scientists from multiple southern African countries across academia and government, have started such an initiative to aid the conservation of intraspecific biodiversity. The initiative relies on open access to DSI from many African countries and will increasingly rely on DSI as it grows and expands to more countries across the continent. If the genetic composition of African species, most of which have multi-national natural distributions, is to be adequately conserved, the

**2200032 (2 of 3)**

continued availability of DSI as a shared resource across the continent is essential. Furthermore, as many African countries have limited resources to generate new DSI, access to existing DSI will allow for collaborations and networks, such as the African node of the IUCN Conservation Genetics Specialist Group, to enhance research capacity and conservation outcomes across the continent.

## Supporting Information

Supporting Information is available from the Wiley Online Library or from the author.

## Acknowledgements

I.M.R. and D.d.J. contributed equally to this work. Funding was provided to I.M.R. by the Higher Education Funding Council for Wales (HEFCW) under the Global Challenges Research Fund (Global Challenges Research Fund; JA1910RD25) program. D.d.J. has received funding from the European Union's Horizon 2020 research and innovation programme under the Marie Sklodowska-Curie grant agreement (101026951). Data sharing is not applicable to this article as no new data were created or analyzed in this study.

## Conflict of Interest

The authors declare no conflict of interest.

## Peer Review

The peer review history for this article is available in the Supporting Information for this article.

## Keywords

conservation, genetic diversity, policy, public databases, translocation

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
