## [**Supplementary Information**: Record of Transparent Peer Review · Advanced Genetics]

The contribution of Digital Sequence Information to conservation biology: a southern African perspective

Isa-Rita M. Russo, Deon de Jager, Anna M. van Wyk, Arrie W. Klopper, Kenneth Uiseb, Coral Birss, Ian Rushworth, Paulette Bloomer

Date submitted: 22 November 2022

Editors: Cate Livingstone, Andrew L. Hufton

1 st Peer Review Decision	16 December 2022
------------------

Dear Isa-Rita,

Thank you for submitting your Comment manuscript entitled "The contribution of Digital Sequencing Information (DSI) to conservation biology: a southern African perspective" (Comment, No. ggn2.202200032) to Advanced Genetics. Your piece has now been reviewed by myself and an additional expert, whose comments are included at the end of this email. I have also attached a Word document with some additional minor comments and edits from myself.

As you will see, the expert we consulted was generally quite positive about this piece and the overall importance of the topic, and recommended publication.

We would therefore invite you to prepare a final version of this manuscript, taking into account the key points listed below:

1. Since it will not be possible to publish this paper before the end of the COP15 biodiversity negotiations, taking place now in Montreal, we would encourage you to wait for the outcomes of this meeting and then revise the text appropriately. It seems extremely unlikely that the DSI issue will be fully resolved, but it remains possible that there could be some developments worth incorporating into your piece. For example, if there is an agreement to develop a multilateral solution for DSI, you may wish acknowledge that agreement in your paper. Please note that the journal will be publishing an Editorial in the next few days on the DSI issue and the COP15 negotiations. Please do not feel under any obligation at all to cite this Editorial, and, indeed, we would encourage you to be sparing with the final number of references.

2. Please include a brief definition of DSI somewhere in the piece, since it is not a common term to many geneticists. I would suggest something brief like: "Digital Sequence Information (DSI), a policy term that encompasses genetic data and related information, ..."

3. Please note the comment from Reviewer #1, and consider whether it might perhaps be useful to add one additional paragraph discussing in a bit more detail the potential impacts of species translocations.

4. While considering the suggestions above, please aim to preserve the concise and clear format and tone of the current piece. Please do not include more than 25 references in the final piece and do not let the length grow substantially (one or two additional paragraphs would be fine, if needed).

To submit your revision, go to <https://www.editorialmanager.com/advgenet/> and log in as an Author using your username (*****) and password. Your submission can be found under the menu item "Submissions Needing Revision". The changes to your manuscript should be highlighted in a different color in the primary "Revised Manuscript" file.

In your cover letter, please outline the changes you have made, and upload this as a "Cover Letter to Editor" item. You will also be asked to upload a .zip archive containing the production data that will be used if your manuscript is accepted. See below for more details.

We should receive your revised manuscript by 22 Dec 2022. Once we receive your revised manuscript, we will provide a final decision as soon as possible.

I would also like to inform you that I will be changing positions at the end of this year, and therefore will likely not be leading the journal when you resubmit. My colleague Lei Lei will be taking on the Editor-in-Chief for the journal. I have informed her of this submission, and am sure it will be handled in a timely manner.

For any future queries, for example regarding the research paper you are considering submitting to the journal, I would recommend writing to the general journal inbox in the first instance (AdvGenet@wiley.com).

Best regards,

Andrew

--

Andrew L Hufton, PhD, Editor
Advanced Genetics
E-mail: AdvGenet@wiley.com
Tel: +49(0)6201-606-362

<http://www.advgenet.com>

REVIEWER REPORT:

COMMENTS TO AUTHOR:

Reviewer #1: Really useful contribution that is aimed specifically at negotiators seeking to develop the DSI approach under the CBD. It elucidates the importance of open access to DSI for wildlife conservation. It points out that artificially enhancing population or species level genetic diversity through translocations needs to be carefully managed, as not all 'increased' genetic diversity is adaptive or desirable (ie. 'diluting' a defined sub-population, sub species or species genomics).

I am not aware of the literature in this field, but while sufficient for a short paper focussed on informing CBD member country negotiators, I was left with the feeling that I need to know more than the theoretical impacts, but also about the genetic and physical impacts to populations through translocations.

Authors' Response to 1 st Review	21 December 2022
---	------------------

21 December 2022

Dr. Andrew L. Hufton,
Editor-in-Chief

Please find attached a revised version of the manuscript (ggn2.202200032) we previously submitted to *Advanced Genetics*: "The contribution of Digital Sequencing Information (DSI) to conservation biology: a southern African perspective". In this Comment we discuss the contribution of DSI to conservation biology with a focus on southern Africa where translocations of wild species are often used as a conservation tool.

We believe that the current version of the manuscript has been improved based on the comments of the reviewers. All changes that have been recommended by reviewers have been addressed and are indicated in red text in the manuscript document. The manuscript has undergone revision and all changes are detailed below. Comments from reviewers are indicated in bold and responses are in red text.

1. Since it will not be possible to publish this paper before the end of the COP15 biodiversity negotiations, taking place now in Montreal, we would encourage you to wait for the outcomes of this meeting and then revise the text appropriately. It seems extremely unlikely that the DSI issue will be fully resolved, but it remains possible that there could be some developments worth incorporating into your piece. For example, if there is an agreement to develop a multilateral solution for DSI, you may wish to acknowledge that agreement in your paper. Please note that the journal will be publishing an Editorial in the next few days on the DSI

issue and the COP15 negotiations. Please do not feel under any obligation at all to cite this Editorial, and, indeed, we would encourage you to be sparing with the final number of references.

We have changed the Abstract. See lines 39-44: “Given the inclusion of specific Goals and Targets regarding DSI in the latest draft of the Global Biodiversity Framework negotiated at COP15 in Montreal in December 2022 and the crucial decisions on access and benefit sharing to DSI that will be taken in the coming months and future COP meetings, we provide a southern African perspective on how and why open access to DSI is essential for the conservation of intraspecific biodiversity (genetic diversity and structure) across country borders.”

In addition to the above changes to the Abstract we have included a paragraph describing the decisions and recommendations that have been recently published. See lines 137-149: “Goal C in the latest Kunming-Montreal Global Biodiversity Framework (Draft decision submitted by the President, CBD/COP/15/L.25, published on 18 December 2022) states “The monetary and nonmonetary benefits from the utilization of genetic resources, and digital sequence information on genetic resources, ... are shared fairly and equitably”, while Target 13 actions “Take effective legal, policy, administrative and capacity-building measures...” to implement Goal C. [22] Furthermore, paragraph 16 of the document CBD/COP/15/L.30 (Digital sequence information on genetic resources) states that The Conference of the Parties “Decides to establish, as part of the post-2020 global biodiversity framework, a multilateral mechanism for benefit-sharing from the use of digital sequence information on genetic resources, including a global fund”, and in paragraph 9 states that the mechanism should “Not hinder research and innovation” and “Be consistent with open access to [23] We support this multilateral, open access and benefit-sharing option for DSI under the Kunming-Montreal Global Biodiversity Framework.[24]

2. Please include a brief definition of DSI somewhere in the piece since it is not a common term to many geneticists. I would suggest something brief like: “Digital Sequence Information (DSI), a policy term that encompasses genetic data and related information, ...”

We have included a definition for DSI in lines 82-85: “This is despite the availability of population and conservation genetics studies and Digital Sequence Information (DSI), which is a policy term that refers to digitally stored information from DNA and RNA (i.e., genetic sequences).”

3. Please note the comment from Reviewer #1, and consider whether it might perhaps be useful to add one additional paragraph discussing in a bit more detail the potential impacts of species translocations.

We have added a sentence (lines 47-48) to emphasise the importance of genetic diversity: “Genetic diversity is essential to evolutionary processes and is important in ecosystem stability, while low genetic diversity increases the extinction risk of populations.[2,3]”.

An additional paragraph has been included in the text to describe the potential impacts of species translocations. Lines 60-65: “The goal of translocations (moving individuals from one population to another) is often to maintain and/or increase genetic diversity in the receiving population. This allows connectivity (gene flow) between populations that have been isolated by human activities, such as land-use change, urbanisation, and fences. Genetic diversity allows populations to adapt to

changing conditions over time, including disease outbreaks and environmental changes.”

4. While considering the suggestions above, please aim to preserve the concise and clear format and tone of the current piece. Please do not include more than 25 references in the final piece and do not let the length grow substantially (one or two additional paragraphs would be fine, if needed).

We have not included more than 25 references, but we did add 3 additional references with regards to the latest CBD documents and we have only included two extra paragraphs.

The minor edits/comments by the Editor-in-Chief have also been addressed. See red text in manuscript document.

We trust you will find all changes satisfactory, and we are looking forward to hearing from you in the future.

Yours sincerely,
Isa-Rita Russo
(On behalf of co-authors)

Final Decision	10 February 2022
----------------	------------------

Dear Dr Russo,

Thank you for submitting your Comment entitled "The contribution of Digital Sequence Information to conservation biology: a southern African perspective" (Comment, No. ggn2.202200032R1) to Advanced Genetics.

I'm sorry for the delay in processing your article but the changes in the editorial team, plus holidays, had an impact but we are back on track now and I am pleased to inform you that your manuscript has been accepted for publication.

We will copyedit the accepted version of your manuscript and if we require any further information at this stage we will contact you. After copyediting we will let you know when you can expect to receive the proofs. Instructions for returning your proof corrections will be provided when the proofs are sent to you.

All articles published in Advanced Genetics are fully open access: immediately and freely available to read, download and share. Advanced Genetics charges a publication fee to cover publication costs. The corresponding author for this manuscript should have already received a quote with the article publication fee, and will soon receive an e-mail invitation to register with or log in to Wiley Author Services (<https://authorservices.wiley.com>). After logging into Wiley Author Services, the publication fee can be paid by credit card, or an invoice or pro forma can be requested. Payment of the publication charge must be received before the article will be published online.

Thank you for choosing Advanced Genetics.

Yours sincerely,

Cate Livingstone

--

Dr Cate Livingstone, Publisher

Advanced Genetics

E-mail: AdvGenet@wiley.com

Tel: +49(0)6201-606-362

<http://www.advgenet.com>